# Perceived Risk and Intentions to Practice Health Protective Behaviors in a Mining-Impacted Region

**DOI:** 10.3390/ijerph17217916

**Published:** 2020-10-28

**Authors:** Courtney M. Cooper, Jeff B. Langman, Dilshani Sarathchandra, Chantal A. Vella, Chloe B. Wardropper

**Affiliations:** 1Water Resources Graduate Program, University of Idaho, Moscow, ID 83844, USA; 2Department of Geological Sciences, University of Idaho, Moscow, ID 83844, USA; jlangman@uidaho.edu; 3Department of Sociology and Anthropology, University of Idaho, Moscow, ID 83844, USA; dilshanis@uidaho.edu; 4Department of Movement Sciences and WWAMI Medical Education Program, University of Idaho, Moscow, ID 83844, USA; cvella@uidaho.edu; 5Department of Natural Resources and Society, University of Idaho, Moscow, ID 83844, USA; cwardropper@uidaho.edu

**Keywords:** Health Belief Model, risk perception, behavioral intentions, lead contamination, mining

## Abstract

Effective risk communication strategies are critical to reducing lead exposure in mining-impacted communities. Understanding the strength of the associations between perceived risk and individuals’ behavioral intentions to protect their health is important for developing these strategies. We conducted a survey within three communities of northern Idaho, USA (*n* = 306) in or near a Superfund Megasite with legacy mining contamination. Survey data were used to test a theoretical model based on the Health Belief Model. Respondents had higher intentions to practice health protective behaviors when they perceived the risk of lead contamination as severe and recognized the benefits of practicing health protective behaviors. Women reported higher behavioral intentions than men, but age and mining affiliation were not significantly associated with behavioral intentions. Although managing lead hazards in communities impacted by mining is challenging due to widely distributed contamination, effective health risk messages, paired with remediation, are powerful tools to protect the health and safety of residents.

## 1. Introduction

Sources of and exposures to lead (Pb), a consequence of industrialization, declined globally following the enforcement of new regulations and its removal from gasoline [1,2]. Despite these improvements, Pb remains a public health hazard and additional precautionary measures are needed [3,4]. Exposure occurs when particles containing Pb are ingested or inhaled [5]. As a dangerous neurotoxin impacting human development, children and pregnant women are especially vulnerable to exposure [6]. However, the toxin impacts every organ of the body and even low levels of exposure are linked to chronic diseases such as diabetes and heart disease [5,7,8,9]. In fact, health organizations including the US Centers for Disease Control (CDC) and the European Food Safety Authority have concluded that there is no safe or “nontoxic” blood lead level [5,10,11].

Managing Pb hazards in communities impacted by mining and smelting is challenging as contaminants are widely distributed in topsoil and dust [3]. In these communities, efforts to remove or remediate Pb often do not eliminate the hazard completely [12]. Instead, the most effective way to reduce exposure is by avoiding any potentially contaminated areas. Unfortunately, sources of exposure are not easily detected and areas that appear clean or safe can still be hazardous [13]. For instance, Pb contaminants may be present in sand at beaches downstream from former or current mine waste discharges [14]. Another possible source of exposure is through the inhalation of house dust that has been contaminated with Pb particles carried indoors on shoes and clothing [15]. Individual health protective behaviors, such as handwashing and leaving potentially contaminated gear outdoors, are important to reducing the risk of exposure in mining-impacted communities [16,17].

In cases where individual health protective behaviors are needed, risk communication strategies and education campaigns that encourage those behaviors are an important form of risk reduction [18,19]. In mining-impacted communities, health risk messages are used to remind people about Pb hazards and the importance of practicing health protective behaviors. For instance, health risk messages are posted at potentially contaminated recreation sites to warn people to take precautions [20]. An individual’s acceptance of a health risk message is influenced by their beliefs both about the health risk and the recommended health behaviors [19]. Previous research suggests that the efficacy of health risk messages is improved when messages are tailored based on knowledge of the relationship between beliefs and behaviors [21]. Identifying associations between health beliefs and health protective behavioral intentions can aid responsible agencies in developing or modifying health risk messages in mining-impacted communities.

Developing consistent and comparable empirical methods for measuring health beliefs requires establishing appropriate scales that can be tailored to specific contexts. Dozens of frameworks are available to guide efforts to understand health beliefs and behaviors (e.g., [22,23]). One of these frameworks, the Health Belief Model (HBM), has guided previous empirical studies related to environmental health risks such as groundwater contamination and well water testing behaviors (e.g., [24,25]). The extended HBM includes six health belief constructs hypothesized to be associated with health behavioral intentions [23]. Health behavioral intentions are considered the most reliable measure of actual behavior [26]. In this study, we used the HBM because, relative to other health behavior frameworks, it is simple to employ in new contexts and because it has guided previous studies about environmental health risks.

Our objective was to examine the strength of associations between health beliefs and health protective behavioral intentions (hereafter, behavioral intentions) by conducting a drop-off, pick-up survey of households in a mining-impacted county in northern Idaho, USA, where widespread Pb contamination has been present for over 140 years [27]. The surveyed communities are located in an active designated Superfund site—the US program administered by the Environmental Protection Agency (EPA) for remediating contaminated areas. In 1983, residential areas were included in the boundaries of the Bunker Hill Superfund site. In 2002, the site boundaries were expanded to include all communities along the floodplains in the region [28]. As of 2018, staff with the Panhandle Health District (District) were concerned that peoples’ behaviors, such as recreating at old mine sites, were leading to Pb exposure [29]. To improve the District’s risk communication strategy, we partnered with the District to develop a survey examining health beliefs and behavioral intentions given possible exposure to Pb.

## 2. Theoretical Framework and Hypotheses

Originally, the HBM was established to expand the use of sociopsychological constructs to determine what factors influence protective health behavior [23,30]. Four of the HBM constructs—perceived severity, perceived susceptibility, perceived barriers, perceived benefits—were developed in the original HBM, while the cue to action and self-efficacy constructs were added to the extended HBM [30]. Perceived severity and susceptibility are defined as an individual’s cognitive assessment of the likelihood and magnitude of a danger [23]. Perceived barriers and benefits relate to individuals’ expectations about the likelihood that an action will be followed by particular consequences [31]. Self-efficacy was added to the HBM because a disbelief in one’s ability to practice a behavior is also believed to influence behavior [32]. Similarly, the cue to action construct—internal and external triggers that prompt a health behavior—was added to the model because of the importance of reminders in facilitating behavior change [23]. Collectively, the six constructs inform individuals’ perceived risk. Motivation to practice health protective behaviors is also influenced by demographic and psychosocial factors that affect an individual’s risk perception [33].

Previous reviews of empirical HBM studies indicate that the associations between the perceived health risk constructs and behavioral intentions are not the same for all the constructs. Meta-analyses of empirical HBM studies find that perceived barriers and benefits tend to be the strongest predictor of behavior [34]. Although not included in many HBM studies, self-efficacy is a strong predictor of behavioral intentions [35]. Self-efficacy is at times considered a perceived barrier, rather than a separate component of the model because, if an individual’s belief in their ability to change their behavior is low because of low self-efficacy, then these beliefs could be interpreted as a perceived barrier. Meta-analyses provide limited insight about the strength of associations between the HBM constructs and behavioral intentions because associations depend on the context and behavior being examined [36,37]. Thus, although the associations proposed by the HBM are reinforced by empirical studies, evaluations of health beliefs and behaviors through the model must be reevaluated across contexts and behaviors.

### Hypotheses

The HBM can be used to evaluate a causal structure, or parallel mediation model, in which an independent variable such as behavioral intention is associated with all of the HBM constructs, and then these affect an outcome variable or an actual behavior [37]. The HBM framework does not assume shared influence or paths between constructs [23]. We proposed associations among HBM constructs and behavioral intentions that can be evaluated by a parallel mediation (path) modeling approach. We hypothesized that the six HBM constructs are associated with behavioral intentions for practicing protective health behaviors in this environment of long-term Pb contamination. For the perceived benefits and barriers constructs, we hypothesized that high perceived benefits and low perceived barriers to action will be associated with behavioral intentions.

Age, gender, and mining affiliation were included in the analysis as covariates because they have been found to influence health protective behaviors in the context of Pb contamination. Age may play a role because a person’s age is an indicator of how much experience he/she has with Pb contamination and whether they are likely to have young children [38,39]. Gender may influence behavior because women and children are more vulnerable to negative health outcomes from Pb exposure and might be more likely to practice health behaviors [40,41]. Affiliation with mining is likely relevant because several previous studies have linked involvement in livelihoods related to a polluting industry with lower perceived health risk [42,43].

## 3. Materials and Methods

### 3.1. Study Area

The study area included three communities in Shoshone County Idaho, USA—Pinehurst, Kellogg, and Wallace (Figure 1). Historical mining, smelting, and associated waste disposal practices resulted in the contamination of soils, sediments, groundwater, and surface water with Pb as well as arsenic and other toxic metals [27]. The county has an aging population of approximately 12,700 (U.S. Census, 2018). The population is predominantly white and poorer than most counties in Idaho [44]. Over 20% of the county’s population under the age of 65 years is on disability, compared to 13% for the state of Idaho. Nearly 9% of children (under 18 years old) have a disability, relative to just over 4% of children in the state [45]. The county also reports higher rates of noncommunicable diseases (e.g., cardiovascular disease) relative to the state [46]. 

Land remediation, through soil removal and replacement of city infrastructure, began in the 1980s following the designation of the Superfund site [49]. Today, most residential properties have been remediated along with the smelting areas and some of the old mine sites [15,48]. Blood lead level concentrations among children living in nearby communities fell from approximately 64 to 2.7 μg/dL during 1974–2001 [11,50]. Yet, the region remains contaminated at abandoned mine sites and in the floodplains of creeks and rivers where the mine waste was dumped and continues to be distributed by high water events [28,51]. With the remediation of most residential properties, Pb exposure now occurs primarily through use of recreational areas in floodplains and near mine sites [15,48].

### 3.2. Survey Development

To assess health beliefs, based on the constructs of the HBM, and behavioral intentions of residents, a drop-off, pick-up survey was developed by the University of Idaho researchers and District staff. The study protocol was approved and certified exempt by the University of Idaho Institutional Review Board (#18-080). Survey questions were developed with guidance from empirical HBM studies. The behavioral intentions were based on the District’s risk communication strategies.

#### 3.2.1. Survey Measures

We used a 5-point response scale for the 33 survey items conceptualized to measure the study variables—the six health belief constructs and a behavioral intentions variable. We chose a 5-point scale because some studies suggest that it offers higher data quality than 7- or 11-point scales [52], and we were concerned about overwhelming respondents with too many response options. The full survey instrument is included in the Appendix A, but not all survey items were used in this analysis. The survey items used in the analysis included:Behavioral intentions: Respondents were asked to consider their intentions to complete six health protective behaviors related to avoiding exposure to Pb contamination over the next year. These behaviors included: removing dirt from clothes, toys, pets, cars, and equipment after spending time outdoors; staying on designated trails while recreating in areas with lead contamination warning signs; washing hands with clean water or wipes before eating or drinking after recreating or working outdoors; using a protective barrier such as a blanket when sitting on a sandy beach; and following the advice of a local public health official about ways to safely avoid lead contamination. Participants responded to all items on a 1 = very unlikely to 5 = very likely scale. A “does not apply” option was included, but these responses were excluded from the analysis. One example of an item measuring behavioral intentions asked about the likelihood that the participant would, “promptly removing dirt from your clothes, toys, pets, cars, and equipment after spending time outdoors.”HBM constructs: Twenty-seven items were included in the survey to measure the HBM constructs perceived severity, perceived susceptibility, perceived benefits, perceived barriers, and self-efficacy. Participants responded to all items on a 1 = strongly disagree to 5 = strongly agree scale. These items were adopted from several studies related to heavy metal contamination (e.g., [16,25,39,53]. The cue to action construct was measured through two items that asked respondents about how frequently they had thought about, read, or heard about Pb contamination issues in the past year. The scale ranged from 1 = never to 5 = very often. One example item, a measure for the perceived susceptibility construct, was, “I have experienced health effects related to lead contamination.”Sociodemographic characteristics: Eight items were about sociodemographic characteristics. Three items were included in the final analysis as covariates due to their possible influence on behavioral intentions. Survey respondents were asked to indicate gender (male, female, and prefer not to answer), age (continuous), and connections to mining. The latter item was phrased: “has a member of your household ever worked in a mining-related job in your local area?” Response options included “yes,” “no,” and “I do not know.” Response options for “I do not know” and “prefer not to answer” were excluded from the model analysis. Remaining items about the sociodemographic characteristics of the sample are reported in the results.

#### 3.2.2. Survey Pretesting

The initial survey was reviewed by the District, five experts in environmental health risk analysis, and a group of nonexperts, including 45 participants solicited through Amazon’s Mechanical Turk. Feedback from this initial survey review informed several revisions to the survey questions to improve clarity. The survey was then pretested at community events hosted by the District in 2018 (*n* = 87). The events included venues where the District regularly conducts education and outreach, such as the North Idaho Agriculture Fair. We elected to pretest the survey only at events that took place in neighboring counties to minimize the possibility of surveying potential survey respondents during pretesting. Results from survey pretesting were submitted to a principal components analysis (varimax rotation) to determine whether the survey items aligned with the HBM constructs. The analysis of the pretest survey responses was deemed acceptable because alignment between survey items and the corresponding HBM variables produced factor loadings greater than 0.4 [54].

#### 3.2.3. Drop-off, Pick-up Survey Procedures

To conduct the drop-off, pick-up survey, we drew stratified random samples from neighborhood clusters in Kellogg, Pinehurst, and Wallace (Table 1). These three communities were chosen from the seven communities in the study area based on their variable locations within the Superfund site, differing population sizes, and ease of access to researchers, as we determined through consultation with the District and the 2010 U.S. Census data. The samples were stratified based on proportional representation of single- and multifamily housing in each community. Neighborhood clusters increased the efficiency of house-to-house visits. A total of 773 households were identified for inclusion in the study. The DOPU method was selected because of its suitability for limiting nonresponse bias within hard to reach communities [55]. The survey was fielded in March 2019. Conducting the survey in March helped to ensure that the sample primarily included our target respondents—residents in the study area communities—rather than winter and summer second home and rental populations.

A presurvey notification was mailed a week prior to the in-person survey drop-off period. Then, the field staff visited each household up to three times to deliver surveys. Only consenting adults (18 years of age or older) were eligible to participate, and participation was randomized by requesting the responsible adult with the closest birthday complete the survey [56]. When a respondent agreed to participate, field staff left the survey and a pen and coordinated a time to return to the house to collect the completed survey. After three failed delivery attempts, field staff left a survey packet (including a cover letter, survey, pen, and prepaid return envelope) on the door of the residence. Overall, 306 surveys were completed, with 30 of those completed by mail. Of the 773 households in the original sample, 204 were excluded because they were determined to be either vacant or unsafe; thus, the final response rate, calculated out of a possible 569 households, was 53.8%. Completed survey data were manually entered into Qualtrics, an online survey platform. Each survey was entered twice by two different researchers and an accuracy check was performed. Discrepancies between the two entries (<1%) were manually corrected.

### 3.3. Data Analysis

RStudio (version 1.2.1335) (PBC, Boston, MA, USA) was used to analyze the data. Descriptive statistics (means, frequencies, and standard deviations) were calculated for the Likert-type questions and the demographic variables. Analysis included structural equation modeling (SEM), a combination of factor analysis and multiple regression analysis [57], which allowed for evaluation of the structural relationship between items in the survey and the latent variables of the HBM.

#### 3.3.1. Multiple Imputation

The data were analyzed for outliers and missing items—survey responses with fewer than 25% missing items, which resulted in excluding 18 incomplete surveys. Although many methods exist for handling missing data within SEM, multiple imputation or maximum likelihood methods are recommended because they produce consistent parameter estimates, standard errors, and test statistics [58]. Prior to the statistical modeling analysis, we conducted 20 rounds of multiple imputation using a maximum likelihood estimation in the R package mice: Multivariate Imputation by Chained Equations for the remaining missing response items (<1% of responses), including for responses coded as “does not apply” [59].

#### 3.3.2. Structural Equation Model Analysis

Model testing was performed in the R package lavaan [60]. Analysis included an exploratory factor analysis (EFA) followed by a two-step SEM to evaluate the measurement and structural properties of modeled associations between the HBM constructs and behavioral intentions. Data were first assessed for factorability using Bartlett’s test of sphericity and the Kaiser–Meyer–Olkin measure (KMO) of sampling adequacy [61]. The Bartlett’s test of sphericity resulted in a Pearson’s chi-square test statistic, χ² (465, *n* = 306) = 5466, *p* < 0.001, and a KMO value = 0.83 above the acceptable threshold of 0.5. Because of the significance of the Bartlett’s test and the KMO value being in the acceptable range, data were deemed suitable for factor analysis. An EFA was performed to evaluate the correlation of the items, validate their groupings with the theorized health belief constructs and behavioral intentions, and establish variable parameters for modeling [54]. The EFA was performed using maximum likelihood extraction method with a direct oblimin rotation due to the expected correlation between the survey items [62]. Factor selection was performed using the Kaiser–Guttman rule based on an eigenvalue cutoff of one [54]. As a preliminary test of validity, each factor was analyzed for internal reliability using Cronbach’s alpha and a threshold of ≥0.7 [63].

The variable groupings derived from the EFA were entered into a confirmatory factor analysis (CFA) to evaluate the measurement properties of the SEM [64]. The factors (variables) identified in the EFA were treated as latent variables in the CFA. The model was estimated with a weighted least square mean and variance (WLSMV) adjusted estimator as a suitable estimator for ordinal data because it does not assume that the data is normally distributed [65,66,67]. We evaluated model goodness-of-fit to the data using multiple indices and the recommendations in Kline [57]. Given the sensitivity of the chi-squared statistic to sample size, a Comparative Fit Index (CFI), a Tucker Lewis Index (TLI), and a root mean square error of approximation (RMSEA) were used to assess model fit [68]. The acceptable threshold for CFI and TLI is ≥0.9 and for RMSEA is ≤0.08 but ideally below ≤0.05.

After assessing the measurement properties of the SEM, we evaluated its structural properties, or the strength of associations between the perceived health risk and behavioral intentions variables. This evaluation included exploring the correlations between the variables as some studies based on the HBM indicate that variable ordering is important due to possible direct and indirect effects between variables [37]. Error correlation between the perceived health risk variables was allowed. Correlated errors assume that the latent variables share at least one omitted characteristic in common and allow the model to account for the possibility of measurement error that develops from similarly worded and measured items [54,57].

## 4. Results

### 4.1. Sociodemographic Characteristics

Survey respondents were 44% male and 91% white, with an average age of 54.5 (SD = 17.7) years, and 36% of respondents held a bachelor’s degree or higher (Table 2). Overall, the sociodemographic characteristics of the sample were similar to the demographic profile of Shoshone County. However, relative to the Shoshone County population, respondents were more likely to be female and older and had higher educational attainment. On average, respondents reported having lived in the study area for 62% of their lives, with over 75% reporting that they lived in the study area at least 75% of their lives. Household income estimates align closely with the income levels of the study area with 52% reporting annual incomes under $50,000. Ten percent of respondents opted not to provide an estimate of their household income. Slightly fewer than half (44%) of respondents reported having a family member (or being involved themselves) in a mining-related occupation.

### 4.2. Exploratory Factor Analysis

The total variance explained by the EFA was 54% and six factors (variables) were extracted and are included in the Appendix A, Appendix A, which also includes summary statistics for the survey items. Nine items were excluded from further analysis because they did not align with a factor. One theorized behavioral intentions item, “how frequently do you recreate in or near the South Fork of the Coeur d’Alene River?” was excluded because it had a loading of only 0.19. Two items conceptualized as relating to self-efficacy, “I seek out information about lead contamination” and “I can prevent lead contamination from entering my home,” cross-loaded >0.3 with other variables and were therefore removed from the analysis. Two items conceptualized as perceived barriers, “preventing lead contamination from entering my home is inconvenient” and “avoiding lead contamination while spending time outdoors is inconvenient,” and two items conceptualized as self-efficacy, “I can avoid lead contamination while spending time outdoors” and “I can prevent lead contamination from entering my home,” failed to load on any factor. Two other items conceptualized as perceived barriers, “I need more information about how to avoid lead contamination while spending time outdoors,” and “I need more information about how to prevent lead contamination from entering my home,” loaded with perceived severity instead of perceived barriers and were therefore excluded from the analysis.

The items conceptualized to measure self-efficacy were divided into two variables for further analysis. Two items formed a variable of self-efficacy in individual knowledge about Pb contamination—“I know a lot about the health effects from lead contamination” and “I am better informed about the health effects of lead contamination than most people”—while two other items measured self-efficacy in accessing information and resources about Pb contamination—“I know who to ask if I have questions about preventing health effects from lead contamination” and “I am aware of the available resources for preventing health effects of lead contamination.” We named this variable “perceived barriers,” but highlight that these items relate only to barriers about awareness of information and resources. The four items conceptualized to measure perceived barriers did not form a cohesive variable. The final six variables extracted through the EFA included behavioral intentions, self-efficacy, perceived severity, susceptibility, benefits, and barriers. These variables demonstrated acceptable reliability with Cronbach’s alphas above 0.7 [63].

### 4.3. Confirmatory Factor Analysis

The initial model demonstrated acceptable fit (χ^2^ (194, *n* = 306) = 550.32, *p* < 0.001; χ^2^/df = 2.83). Although the chi-square test was significant—suggesting poor model fit—the RMSEA value (0.078) was within the acceptable limits and the CFI (0.987) and TLI (0.985) were above the minimum threshold of 0.9 suggested by Kline [57]. The perceived susceptibility item, “if it is my destiny to experience health effects related to lead contamination, there is nothing that I can do to prevent it,” had a low standardized coefficient of 0.40 relative to the other three items that comprised the perceived susceptibility variable, so the item was dropped from the analysis. High correlations were found between several items and variables. The high correlation between the perceived benefits and perceived severity (r = 0.50, *p* < 0.001) led to a decision to evaluate the structural properties of two models with and without the perceived benefits variable. The full correlation matrix with the six latent variables is given in Appendix A. Following a recommendation by Rosseel [60], we accounted for high correlations in the perceived severity variable items by adding residual variances between items measuring the same cognitive concepts for indoor versus outdoor Pb contamination. The adjusted model revealed improved fit (Table 3: χ^2^ (172, *n* = 306) = 422.30, *p* < 0.001; CFI = 0.992; TLI = 0.990; RMSEA = 0.069). The revised model did not meet the ideal threshold of ≤0.05 for the RMSEA value but was within the acceptable range of ≤0.08. Due to the exploratory nature of the model and acceptable model fit, the revised CFA was considered plausible.

### 4.4. Path Analysis

Table 4 includes the associations between the perceived health risk variables and behavioral intentions based on three path models. Model 1 shows the path coefficients for the variables included in the CFA. Only the association between perceived benefits and behavioral intentions was significant (β = 0.67, *p* < 0.001). Model 2 reflects the same model without the perceived benefits variable (χ^2^ (92, *n* = 306) = 186.76, *p* < 0.001; CFI = 0.976; TLI = 0.994; RMSEA = 0.058). Model 2 had a lower chi-square value and RMSEA than Model 1, and the association between perceived severity and behavioral intentions was significant (β = 0.62, *p* < 0.001). This indicates that perceived severity had a lower effect on behavioral intentions than perceived benefits. 

Model 3 included the latent variables, the two cues to action items, and the covariates gender, age, and mining affiliation (χ^2^ (272, *n* = 306) = 1147.98, *p* < 0.001; CFI = 0.961; TLI = 0.970; RMSEA = 0.103). The path coefficients for the full model are illustrated in Figure 2. The RMSEA exceeded the recommended threshold of 0.08, however, the CFI and TLI were within the acceptable range. Based on discussion in Kline [57] and Xia and Yang [68] we decided that the model fit was reasonable despite the RMSEA because it is sensitive to sample size and was not developed explicitly for ordinal categorical data. The perceived benefits variable was again significantly associated with behavioral intentions (β = 0.61, *p* < 0.001). One of the cues to action variables, the survey item that asked if respondents “thought about lead contamination issues” was significantly associated with behavioral intentions (β = 0.27, *p* < 0.001). Gender was the only covariate with a statistically significant association with behavioral intentions, with women more likely than men to report intentions to practice health protective behaviors (β = −0.36, *p* < 0.001).

## 5. Discussion

According to Model 1, perceived benefits and the cue to action item about how frequently respondents thought about Pb contamination were significantly associated with behavioral intentions, while perceived severity, perceived susceptibility, perceived barriers, and self-efficacy were not. When we excluded the perceived benefits variable from Model 2, due to its high correlation with perceived severity, perceived severity and the same cue to action item had significant associations with behavioral intentions. The results suggest that respondents, who perceived the risk of Pb contamination as severe, believed that there were benefits of protecting themselves from Pb contamination and thought that Pb contamination frequently had higher behavioral intentions. In Model 3, of the covariates tested, only gender had a significant association with behavioral intentions, indicating that women were more likely than men to report intentions to practice health protective behaviors. The next section discusses how exogenous factors related to the Superfund site in the study region may have also influenced the associations between the health belief constructs and behavioral intentions.

### 5.1. Mining-Impacted Communities, Risk Perception, and Behavioral Intentions

Environmental and public health conditions in the study area have improved over the past several decades: hillsides that were once bare due to smoke fallout from the Pb smelter have been revegetated [69], the water quality and condition of rivers and streams have improved [27], health risk warning signs are posted at public recreation areas and at old mining sites [29], the District hosts regular workshops about Pb contamination and provides free annual blood lead screenings [48], and extensive environmental remediation in residential and commercial areas has reduced the risk of Pb exposure [50]. Nationally, the EPA has applied a model of community engagement at Superfund sites in an attempt to ensure that remediation and restoration efforts align with local needs [70]. Related studies demonstrate that risk perception is not only linked to the contaminants present but also to cues such as the equipment used for remediation or how organized remediation sites appear [71]. The improved environmental conditions and focus on health risk communication may cue people to recognize and understand the health benefits to practicing recommended health protective behaviors. 

Although the Superfund program is intended to reduce public health risks and improve living conditions, residents have also developed negative emotional responses towards hazards and the decision-makers associated with the program (e.g., [72,73,74]). Fears of stigmatization and concerns about economic development have long been associated with how people cope with risk in mining-impacted communities [73,75]. Baxter and Lee [76] found that a strong sense of community pride and a fear of stigmatization prevented people from outwardly expressing concern about the health risks of a nearby hazardous waste facility. Grasmück and Scholz [77] reported that the desire for additional information about the risk of heavy metal contamination in soil was not affected by a lack of knowledge but was affected by emotional concerns. Factors like stigmatization and denial may explain why we did not identify significant associations between the HBM constructs for self-efficacy and perceived susceptibility and behavioral intentions. For instance, a person who holds negative emotions towards the Superfund program may report high levels of self-efficacy, but low behavioral intentions due to their negative association with the program making the recommendations.

Perceptions about the health consequences of Pb contamination may also be influenced by environmental remediation and risk communication efforts. In this study, perceived severity, but not perceived susceptibility, was strongly associated with behavioral intentions. The result may be explained by differences in perceptions of acute versus immediate health risks. Walpole and Wilson [78] illustrated that personal perceived risk was significantly associated with perceived severity but not perceived susceptibility for risks related to contaminated waterways, while the inverse was true for more immediate and acute risks such as extreme weather events or walking in a dangerous neighborhood. The existing efforts to abate Pb hazards and a decline in childhood blood lead levels [48] may lead residents to view the negative health consequences of Pb contamination as a long-term rather than acute health issue. If this is the case, individuals may be less likely to attribute Pb contamination with negative health consequences to themselves. 

### 5.2. Sociodemographic Characteristics and Behavioral Intentions

Women were more likely than men to report practicing health protective behaviors. This could be because women and children are more vulnerable to experiencing health effects from Pb contamination [7,16]. Relative to men, women are also often considered more likely to practice health behaviors intended to reduce the consequences of environmental risks [40,79]. The nonsignificant effects for the mining affiliation and age covariates may indicate that they do not influence behavioral intentions relative to the other study variables. Wolde et al. [80] also found that age was not a significant factor informing individual behaviors related to Pb exposure. Considering that children are more sensitive to Pb’s adverse health effects, future studies are needed to determine why younger study participants did not have higher behavioral intentions relative to older participants. Although nearly half of respondents reported having a familial affiliation with the mining industry, mining-related employment opportunities in the study area are limited [27]. Studies where an affiliation with a polluting industry has influenced behavioral intentions have been conducted in areas where the industry plays a more influential economic role than is currently the case in the study area (e.g., [42]).

### 5.3. Study Limitations 

Habitual behaviors such as the behaviors that reduce the risk of Pb exposure are among the more difficult health behaviors to evaluate and monitor. Cross-sectional HBM studies about habitual behaviors have several limitations including: (1) relying on measures of health behavioral intentions variables rather than actual behaviors [26,81]; (2) failing to account for feedbacks between perceived risk and behavior [82,83]; and (3) excluding external cultural, cognitive, or affective responses and biases that may influence risk perception and behavior [81]. The HBM’s benefit in cross-sectional studies is that it is relatively easy to employ and can be applied and compared across contexts and behaviors.

The social desirability bias and the intention-behavior gap were primary limitations. Social desirability bias is defined as a tendency of survey respondents to give responses that they perceive as desired by the researcher instead of choosing responses that are reflective of their true feelings [84]. The DOPU method improved our ability to collect data across a hard to reach population. However, the approach may have increased the influence of social desirability bias as study respondents and researchers interacted with one another directly during data collection. Findings from a meta-analysis of the intention-behavior gap, or the gap between behavioral intentions and behavior, indicate that while intention may be the best predictor of behavior, less than one-third of behavior change can be explained by behavioral intention [26]. We found that survey respondents were “likely” or “very likely” to perform health protective behaviors, yet the District observes residents practicing exposure enhancing behaviors, such as recreating in areas that may have high levels of contamination [29]. These two limitations are not easily resolved within empirical cross-sectional studies.

Finally, the results are limited by the smaller sample size and the mismatch in representation between the survey sample and the study population. Other related studies have reported similar sample sizes (e.g., [25]) and survey respondents are often older, have higher levels of education attainment, and are more likely to be female [56]. A strength of this work is that we did survey respondents in communities that are often considered hard to reach populations.

## 6. Conclusions

The goal of this study was to build understanding of how health beliefs influenced residents’ intentions to practice health protective behaviors. To remind residents about recommended health protective behaviors, the District could send an annual reminder postcard to residents. Because young children are especially vulnerable to Pb poisoning and we found that age did not have a statistically significant association with behavioral intentions, risk communication strategies could focus on approaches that encourage parents to practice behaviors that limit Pb exposure for their children. For instance, the District could work directly with pediatricians to share information about Pb poisoning with parents.

Results from this study are transferable to other contexts where frequent long-term exposure to contaminants is linked to long-term health consequences and individual behaviors are important for reducing exposure. Results are likely to be most similar in contexts with developed programs for managing Pb contamination. The HBM provides a useful yet incomplete conceptual framework for understanding interactions between behavioral intentions and health beliefs. Ongoing remediation and restoration activities, changes to mining operations, prolonged psychological effects of stigmatization, and reductions in childhood blood lead levels likely also influenced the HBM constructs. Methods for evaluating these factors should be a focus for future research.

Comparative methods and longitudinal panel studies are important for understanding dynamic perceptions at Superfund sites and in other contexts with persistent contaminants. Without comparative studies, developing a comprehensive outlook about how risk perception and behavioral intentions are socially determined, versus how they vary between contexts, is not possible. Longitudinal panel studies could help to build understanding of how behavioral intentions and risk perception change over time, especially when management actions such as the remediation or removal of Pb hazards alter the severity of the risk. Continuing to build knowledge about how risk perceptions interact with behavioral intentions is important for developing risk communication strategies tailored to specific contexts and population subgroups. Prioritizing improved risk communication alongside Pb hazard abatement programs is critical for reducing both the societal and individual human health consequences of Pb exposure.

## Figures and Tables

**Figure 1 ijerph-17-07916-f001:**
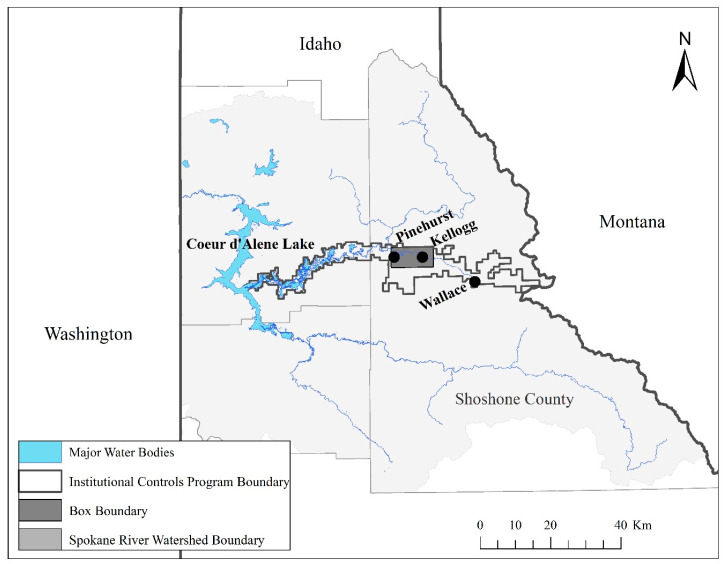
Communities of Kellogg, Pinehurst, and Wallace. The dark gray rectangle incorporating Pinehurst and Kellogg represents the 54 km^2^ area known as “the Box”—the area of the original Bunker Hill Superfund Site that included a smelter and other processing facilities. The Institutional Controls Program Boundary includes the expanded Superfund site that includes 394 km^2^ of floodplains and wetlands. Image produce in Esri ArcGIS 10.8 (Esri, Redlands, CA, USA). Data sources: [47,48].

**Figure 2 ijerph-17-07916-f002:**
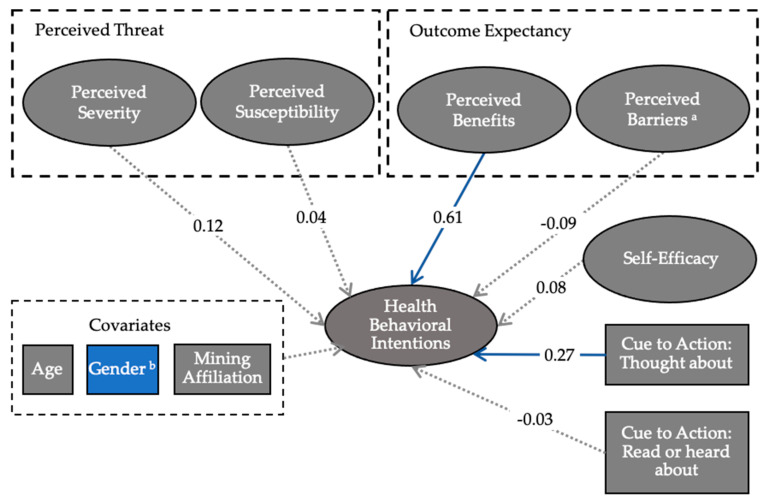
Path analysis for the full model. Solid blue lines indicate significant paths. ^a^ Perceived barriers were hypothesized to be inversely associated with behavioral intentions. ^b^ Gender had a significant association with behavioral intentions in both models with women being more likely than men to report performing health protective behaviors. Ovals represent latent variables and rectangles represent observed variables.

**Table 1 ijerph-17-07916-t001:** Drop-off, pick-up survey responses.

Survey Sample	*n* (%)	Household Type		Community		
		Multifamily	Single-Family	Kellogg	Pinehurst	Wallace
Selected households	773 (100%)	193 (25%)	580 (75%)	365 (47%)	255 (33%)	159 (20%)
Removed from sample						
Vacant/unsafe	204 (26%)					
Refusals	126 (16%)					
Unreturned mailers	119 (15%)					
Incomplete	18 (5%)					
Survey responses	306 (40%)	58 (18%)	248 (82%)	143 (47%)	113 (37%)	49 (16%)

Note: The final analysis was based on 306 surveys. Surveys with more than 20 incomplete items were excluded from the analysis. Towns were sampled proportionately based on number of households.

**Table 2 ijerph-17-07916-t002:** Description of sample (*n* = 306).

Characteristic	Mean (SD) (% (frequency))
Age (years, M (SD))	54.5 (17.7)
Years lived in study region (years, M (SD))	33.3 (21.5)
Gender (% (frequency))	
Female	54% (165)
Male	44% (134)
Prefer not to say	2% (6)
Race/ethnicity (% (frequency))	
White	90.8% (278)
No response	4.6% (14)
All others	5% (14)
Highest education (% (frequency))	
Advanced degree	9.8% (30)
College degree	26.1% (80)
Some college but no degree	30.1% (92)
High school graduate	28.1% (86)
Less than high school degree	5.2% (16)
Occupational status (% (frequency))	
Retired	35.6% (109)
Working full-time	36.3% (114)
Homemaker	8.8% (27)
Working part-time	7.2% (26)
Disabled/medical leave	4.6% (5)
Student	0.7% (2)
Unemployed	1.3% (4)
No response	3.0% (9)
Approximate household income (% (frequency))	
Less than $20,000	21.6% (66)
$20,000 to $49,999	30.7% (94)
$50,000 to $79,999	22.5% (69)
$80,000 to $99,000	8.2% (26)
$100,000 or more	6.5% (21)
No response	10% (30)
Family in mining (% (frequency))	
No	53.3% (163)
Yes	44.4% (136)
Not sure	1.6% (5)

Note: “No response” categories excluded for characteristics when less than 1%.

**Table 3 ijerph-17-07916-t003:** Confirmatory factor analysis of the Health Belief Model (HBM) and behavioral intentions variables (*n* = 306).

Item	*B* (SE) ^a,b^	*β*
Perceived Benefits		
Indicate to what extent you agree that completing the following actions are good for your health:		
Promptly removing dirt from your clothes, toys, pets, cars, and equipment after spending time outdoors.	1.00	0.80
Staying on designated trails while recreating in areas with lead contamination warning signs posted.	1.01 (0.03)	0.87
Washing your hands with clean water or wipes before eating or drinking after recreating or working outdoors.	0.92 (0.04)	0.77
Using a protective barrier such as a blanket when sitting on a sandy beach.	1.03 (0.03)	0.86
Following the advice of a local public health official about ways to safely avoid lead contamination.	0.10 (0.03)	0.83
Perceived Severity		
I worry about lead contamination while spending time outdoors.	1.00	0.80
It is worth my time to avoid lead contamination while spending time outdoors.	1.02 (0.06)	0.79
I worry about lead contamination entering my home.	0.97 (0.04)	0.75
It is worth my time to clean my home to prevent lead contamination.	1.01 (0.06)	0.79
Behavioral Intention		
Consider your recreational and outdoor activities in your local area over the next 12 months. How likely is it that you will?		
Promptly removing dirt from your clothes, toys, pets, cars, and equipment after spending time outdoors.	1.00	0.80
Staying on designated trails while recreating in areas with lead contamination warning signs posted.	0.90 (0.05)	0.72
Washing your hands with clean water or wipes before eating or drinking after recreating or working outdoors.	0.90 (0.06)	0.71
Using a protective barrier such as a blanket when sitting on a sandy beach.	0.97 (0.05)	0.77
Following the advice of a local public health official about ways to safely avoid lead contamination.	1.07 (0.05)	0.85
Perceived Susceptibility		
I have experienced health effects related to lead contamination.	1.00	0.90
I feel I will experience health effects related to lead contamination at some time during my life.	1.10 (0.03)	0.99
I am more likely than the average person to experience health effects from lead contamination.	0.88 (0.03)	0.79
Self-Efficacy		
I know a lot about the health effects from lead contamination.	1.00	0.90
I am better informed about the health effects of lead contamination than most people.	0.96 (0.04)	0.90
Perceived Barriers		
I know who to ask if I have questions about preventing health effects from lead contamination.	1.00	0.90
I am aware of the available resources for preventing health effects of lead contamination.	1.03 (0.03)	0.96

Note: Both unstandardized (b) and standardized (β) beta coefficients are reported. Model: χ^2^ (172, *n* = 306) = 422.30, *p* < 0.001; CFI = 0.992; TLI = 0.990; RMSEA = 0.069). ^a^—The variables’ first items were fixed as reference items at 1.00 in Lavaan. ^b^—Regression weights significant at *p* < 0.001.

**Table 4 ijerph-17-07916-t004:** Associations between HBM variables and behavioral intentions (dependent variable), *n* = 306.

Model	Model 1		Model 2		Model 3	
Independent Variable	*b* (SE)	*β*	*b* (SE)	*β*	*b* (SE)	*β*
Perceived severity	0.17 (0.09)	0.16	0.57 (0.07) *	0.62	0.15 (0.09)	0.12
Perceived susceptibility	0.00 (0.06)	0.00	−0.12 (0.07)	−0.14	0.04 (0.06)	0.04
Perceived benefits	0.64 (0.06) *	0.67			0.63 (0.06) *	0.61
Perceived barriers	−0.06 (0.09)	−0.08	0.07 (0.10)	0.08	−0.08 (0.09)	−0.09
Self-efficacy	0.05 (0.08)	0.06	0.02 (0.09)	0.02	0.08 (0.09)	0.08
^1^ Cue: thought about	0.21 (0.04) *	0.26	0.21 (0.04) *	0.26	0.21 (0.04) *	0.26
^1^ Cue: read or heard about	−0.02 (0.05)	−0.03	−0.02 (0.05)	−0.03	−0.02 (0.05)	−0.03
Gender (0 = F, 1 = M)					−0.36 (0.10) *	−0.22
Mining affiliation (0 = no, 1 = yes)					−0.13 (0.10)	−0.07
Age					−0.00 (0.00)	−0.03

Note: Model 1: χ^2^ (172, *n* = 306) = 422.30, *p* < 0.001; CFI = 0.992; TLI = 0.990; RMSEA = 0.069. Model 2: χ^2^ (92, *n* = 306) = 186.76, *p* < 0.001; CFI = 0.976; TLI = 0.994; RMSEA = 0.058. Model 3: χ^2^ (272, *n* = 306) = 1147.98, *p* < 0.001; CFI = 0.961; TLI = 0.970; RMSEA = 0.103. Both unstandardized (b) and standardized (β) beta coefficients are reported. The coefficients and error terms measure the strength of the statistical association. * associations significant at the *p* < 0.001. ^1^ Cue to action variables based on two survey items.

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
