# Peer review of "Perceived Risk and Intentions to Practice Health Protective Behaviors in a Mining-Impacted Region"

_ijerph, 2020, doi:10.3390/ijerph17217916_

Round 1

Reviewer 1 Report

 This submission is an interesting empirical study that applies the health belief model to communities in northern Idaho. The study has the potential to make a meaningful contribution with interesting results, but the reviewer would like to make some suggestions as below in light of further improving the manuscript.

  1. With regard to the research model, can there be a stronger justification as to why HBM was used for this study? Put differently, is there any advantage/strength of this model against other competing/alternative models?
  2. More details on the results from the survery pretesting is desired.
  3. There could be more justification(s) on why the sample has representativeness of the population.
  4. Please provide more details of the imputation process. Also, was it a multiple imputation?
  5. The sample size may be discussed as a limitation.   

Author Response

With regard to the research model, can there be a stronger justification as to why HBM was used for this study? Put differently, is there any advantage/strength of this model against other competing/alternative models?

Thanks for this comment, the extended HBM was primarily used because of its popularity and descriptive characteristics. We added more discussion about this decision in paragraph 4 of the introduction as well as in paragraph 1 of section 2. We moved the discussion of the theoretical frames and hypotheses out of the methods and into their own separate section (section 2)

More details on the results from the survery pretesting is desired.

We added an additional description about how the survey was pretested (now section 3.2.2). We also want to emphasize that the survey is exploratory. Testing the survey at community events allowed us to ensure that the items were interpretable to the study population. However, future studies should refer to results from the exploratory factor analysis when designing surveys as several items were excluded from our model analysis based on those results.   

There could be more justification(s) on why the sample has representativeness of the population.

We added more discussion of the representativeness of the sample to section 4.1. The survey respondents were older, more female, and more highly educated than the county population. This is a common occurrence in survey research. We made an effort to mitigate the issue by carefully following procedures recommended for conducting drop-off, pick-up surveys. For instance, we visited houses up to three times, at different times of day, before leaving the survey mailer packet at a house. We did this to improve opportunities for making in person contact with respondents recognizing differences in daily work schedules.   

Please provide more details of the imputation process. Also, was it a multiple imputation?

We clarified that the imputation completed was multiple imputation (section 2.3.1). We used 20 rounds of imputation and the maximum likelihood estimator. The amount of missing data was minimal considering norms for drop-off, pick-up surveys. The clarification improves this manuscript. 

The sample size may be discussed as a limitation.   

Thank you for this comment, we agree and have added this as a limitation on lines 478-482. We also note that similar studies have reported similar sample sizes. 

Reviewer 2 Report

Dear authors,

Overall, an interesting paper.  I like the mixture between environmental health and psychology; I think these interdisciplinary studies are the way of the future.  That said, you did not conduct an appropriate qualitative analysis and yet, you have qualitative data in this paper.  Please remove this or rewrite the paper to confirm it’s mixed methodology.  I think you did triangulation, but this needs to be abundantly clear in your methods section.  When I write and publish my mixed methods papers, I differentiate each section “qualitative design” and “quantitative design” and then further within each group “collection”, “analysis” etc.  When I review results, I also separate them.  This makes it so that I am extremely transparent about both methodological processes.  I suggest you review this chapter https://www.sagepub.com/sites/default/files/upm-binaries/10982_Chapter_4.pdf Which is helpful.  OR, you could delete the comments section in here (that would be the easiest to do) and focus just on your quantitative results.  That, for me, would significantly improve the paper.

Abstract

Rewrite first sentence of abstract.  It’s confusing and long.  Add one sentence about exposure to lead, in general, and a second sentence about education and knowledge and behavior change.

Intro

Tons of commas are missing, which makes it difficult to read.

Last sentence, I would state that though a site has been cleaned or remediated, hazards may still exist. Then, the next paragraph needs a transition sentence from this statement to how human populations are at risk.  You have that, you just don’t have a lead in sentence.

Why would it reduce crime and increase academic achievement?

Line 48, add the example, not just the reference points.

Link the last paragraph back to the risks again that lie in these communities surrounding this site.

Methods

Add that Silver County is in Idaho, USA in the FIRST sentence.

Can you cite this extremely important sentence? “Yet, the Silver 91 Valley remains contaminated at abandoned mine sites and in the floodplains of creeks and rivers 92 where the mine waste was dumped and continues to be distributed by high flows.”

Survey questions, but what type of study was conducted?

I’m still reading about the survey and have no idea if this is mixed methods, qualitative, or quantitative.  What types of questions were on it?  How many?  Add more information.

You can’t just have comments at the end of a survey.  That is called qualitative research.  That needs to be aligned with a mixed methodological study with encompassing quantitative and qualitative research paradigms.  Both of these methodologies need to have an entire section on data collection, analysis, participant demographic data (for those who completed it), how you triangulated data through questions, and how you analyzed it.  I would seriously consider deleting this section, or add some significant content regarding this qualitative data collection and analysis.  This is an automatic manuscript reject for me, if unchanged.

Discussion

Delete the comments paragraph.

Don’t say, the Silver Valley.  People in Africa, Asia, etc. don’t know what you are talking about… Focus on the remediated site that has contamination, which just happens to be in this location.

I would also say, theoretically supported by the Health Belief Model.  This is a major psych theory.

Author Response

Overall, an interesting paper.  I like the mixture between environmental health and psychology; I think these interdisciplinary studies are the way of the future.  That said, you did not conduct an appropriate qualitative analysis and yet, you have qualitative data in this paper.  Please remove this or rewrite the paper to confirm it’s mixed methodology.  I think you did triangulation, but this needs to be abundantly clear in your methods section.  When I write and publish my mixed methods papers, I differentiate each section “qualitative design” and “quantitative design” and then further within each group “collection”, “analysis” etc.  When I review results, I also separate them.  This makes it so that I am extremely transparent about both methodological processes.  I suggest you review this chapter https://www.sagepub.com/sites/default/files/upm-binaries/10982_Chapter_4.pdf Which is helpful.  OR, you could delete the comments section in here (that would be the easiest to do) and focus just on your quantitative results.  That, for me, would significantly improve the paper.

As suggested, we deleted the sections related to the survey comments throughout as well as associated discussion points. We understand that to include the analysis about the comments we should also comprehensively describe the qualitative methodology. Thanks for your interest in interdisciplinary studies!

Abstract

Rewrite first sentence of abstract.  It’s confusing and long.  Add one sentence about exposure to lead, in general, and a second sentence about education and knowledge and behavior change.

We split the first sentence of the abstract in two sentences as suggested. This comment strengthens the paper abstract. 

Intro

Tons of commas are missing, which makes it difficult to read.

We rewrote the introduction and focused on improving the readability. We also carefully reviewed the grammar.

Last sentence, I would state that though a site has been cleaned or remediated, hazards may still exist. Then, the next paragraph needs a transition sentence from this statement to how human populations are at risk.  You have that, you just don’t have a lead in sentence.

This is a great suggestion. We reorganized the introduction and focused on improving our description of how human populations remain at risk even after remediation. This description is now in paragraph 2. We also focused more on defining the health protective behaviors that are associated with reduced exposure in paragraph two.  

Why would it reduce crime and increase academic achievement?

We deleted the citation as it seems to distract from the introduction purpose. We refocused on describing the health consequences of lead poisoning more directly. See paragraph 1. We refocused the introduction to better clarify the background terminology most relevant for developing our research objectives. 

Line 48, add the example, not just the reference points.

We deleted this reference in the revised introduction. 

Link the last paragraph back to the risks again that lie in these communities surrounding this site.

Thanks for this comment. We added this description in lines 60-62

We rewrote the introduction to include a clearer description of relevant terminology in the study. 

Methods

Add that Silver County is in Idaho, USA in the FIRST sentence.

We made this change, see line 77. We also clarified the study location in the objective paragraph line 63-65. We also deleted all instances of the “Silver Valley” as suggested below. 

Can you cite this extremely important sentence? “Yet, the Silver 91 Valley remains contaminated at abandoned mine sites and in the floodplains of creeks and rivers 92 where the mine waste was dumped and continues to be distributed by high flows.”

Thanks for pointing out this important omission. We add two citations for this sentence. The citations focus on both the floodplains and mine sites (line 146-147). 

Survey questions, but what type of study was conducted?

We added to our objective paragraph lines 69-80 that we conducted a household survey. Eliminating the survey comments also clarifies the kind of study that was conducted. We did not intend to develop a mixed methodology paper. The entire survey is included in the supplementary materials.  

I’m still reading about the survey and have no idea if this is mixed methods, qualitative, or quantitative.  What types of questions were on it?  How many?  Add more information.

Thank you for these comments. We reorganized what is now section 3.2 to better describe the process of developing the survey. Descriptions about the survey questions (items) are included in section 3.2.1. Table 3 also lists the survey items that were included in the model. We added examples of the survey questions to this section as well. The full survey and the exploratory factor analysis and summary statistics for the survey items are in the supplementary materials. We also reorganized the paper, making a separate section (now section 2) to describe the theoretical frame and hypotheses rather than including the description in the methods section.  We believe these clarifications and the reorganization helped to strengthen the manuscript. 

You can’t just have comments at the end of a survey.  That is called qualitative research.  That needs to be aligned with a mixed methodological study with encompassing quantitative and qualitative research paradigms.  Both of these methodologies need to have an entire section on data collection, analysis, participant demographic data (for those who completed it), how you triangulated data through questions, and how you analyzed it.  I would seriously consider deleting this section, or add some significant content regarding this qualitative data collection and analysis.  This is an automatic manuscript reject for me, if unchanged.

Thank you for this comment. We deleted the analysis about the survey comments throughout the document. 

Discussion

Delete the comments paragraph.

We deleted this paragraph as noted above. 

Don’t say, the Silver Valley.  People in Africa, Asia, etc. don’t know what you are talking about… Focus on the remediated site that has contamination, which just happens to be in this location.

We deleted all instances of the Silver Valley throughout the manuscript and instead refer to the location as the study area. 

I would also say, theoretically supported by the Health Belief Model.  This is a major psych theory.

Thank you, the HBM is a useful framework for evaluating health beliefs and behavioral intentions. We hope that future research will continue to build from this study through comparative and longitudinal research. 

Reviewer 3 Report

In general the study is well conducted, relevant to the readers of the journal, and while I am a different academic background, it is written for a broad audience. Therefore, this study is suitable for the readers of this journal. I have only a few minor comments, which the authors should consider:

Line 38: development issues such as?

Line 43: The study of Muennig [16] states that reducing blood lead levels among children results in reduced crime. I am in a somewhat different field, but maybe a bit more explanation is required here (how does crime related to lead levels?).

The quality of figure 1 can be improved.

Line 49-55: It would be could to strengthen the rationale why the HBM was chosen. This section is relatively short, and could be further elaborated on.

The authors could opt to write about the development of survey development including respondent statements, in the supplementary materials.

Author Response

In general the study is well conducted, relevant to the readers of the journal, and while I am a different academic background, it is written for a broad audience. Therefore, this study is suitable for the readers of this journal. I have only a few minor comments, which the authors should consider:

Thank you, we appreciate your review of our work. 

Line 38: development issues such as?

We deleted this part of the introduction as it was primarily tied to the section about survey comments. Another reviewer requested that we remove this section. We believe that the deletion improved the manuscript. 

Line 43: The study of Muennig [16] states that reducing blood lead levels among children results in reduced crime. I am in a somewhat different field, but maybe a bit more explanation is required here (how does crime related to lead levels?).

An excellent point, however, based on other reviewer comments and the revisions to the introduction, we removed this citation. Other studies have connected childhood lead poisoning to increased crime and high school dropout rates. We focused the introduction instead on clarifying that lead poisoning causes developmental delays in children. 

The quality of figure 1 can be improved.

We improved the quality of figure 1. 

Line 49-55: It would be could to strengthen the rationale why the HBM was chosen. This section is relatively short, and could be further elaborated on.

We built on the rationale both in the introduction (paragraph 4). We also reorganized the paper and made a separate section to discuss model choice and hypotheses (now section 2). 

The authors could opt to write about the development of survey development including respondent statements, in the supplementary materials.

Thanks for making this suggestion. To avoid making the paper appear to draw from mixed methodologies, we removed the survey comments. Deleting the survey comments makes the paper clearer and easier to follow. 

Round 2

Reviewer 2 Report

Significantly improved.  Thank you for deleting the qualitative aspect of this project.  I recommend publication.